# Knowledge and Myths about Palliative Care among the General Public and Health Care Professionals in Portugal

**DOI:** 10.3390/ijerph17134630

**Published:** 2020-06-27

**Authors:** Maria dos Anjos Dixe, Irene Dixe de Oliveira Santo, Saudade Lopes, Helena Catarino, Susana Duarte, Ana Querido, Carlos Laranjeira

**Affiliations:** 1School of Health Sciences of Polytechnic Institute of Leiria (IPLeiria), Campus 2, Morro do Lena, Alto do Vieiro, Apartado 4137, 2411-901 Leiria Leiria, Portugal; saudade.lopes@ipleiria.pt (S.L.); helena.catarino@ipleiria.pt (H.C.); ana.querido@ipleiria.pt (A.Q.); carlos.laranjeira@ipleiria.pt (C.L.); 2Centre for Innovative Care and Health Technology (ciTechCare), Rua de Santo André—66–68, Campus 5, Politécnico de Leiria, 2410-541 Leiria, Portugal; irene.dixe1993@gmail.com; 3Weiss Memorial Hospital, 4646 North Marine Drive, Chicago, IL 60640, USA; 4The Health Sciences Research Unit: Nursing, Nursing School of Coimbra (ESEnfC), 3046-851 Coimbra, Portugal; susanaduarte@esenfc.pt; 5Center for Research in Health and Information Systems (CINTESIS), NursID, University of Porto, 4200-450 Porto, Portugal; 6Research in Education and Community Intervention (RECI I&D), Piaget Institute of Viseu, 3515-776 Viseu, Portugal

**Keywords:** palliative care, knowledge, end-of-life care, health care professionals, nonhealthcare professionals

## Abstract

*Background:* International research has shown that healthcare professionals (HCPs) and nonhealthcare professionals (NHCPs) are unaware of the goals and purposes of palliative care. This study evaluates the knowledge of palliative care among a sample of Portuguese adults and correlates their level of knowledge with age, gender, profession, and experience of family member’s palliative care. *Method:* A cross-sectional online survey was carried out on a sample of 152 HCPs and 440 NHCPs who completed an anonymous questionnaire of sociodemographic, family, and professional data, and an instrument of 26 dichotomous (true or false) questions focusing on palliative care goals and purposes. *Results:* The 592 participants had a mean age of 31.3 ± 11.1 years, and most were female. Statistically significant differences between statements considered as correct by HCPs and NHCPs were found in 24 statements; HCPs had the highest percentage of correct answers. The terms most frequently associated with palliative care mentioned by NHCPs were chronic and progressive disease (n = 76), while HCPs mostly mentioned quality-of-life promotion (n = 29). Women, the elderly, and HCPs had a higher level of knowledge regarding palliative care (*p* < 0.001). *Conclusions*: Results clearly show gaps in knowledge of palliative care, especially among NHCPs. An integrated approach is needed to inform and clarify the philosophy and goals of palliative care in different settings in order to improve knowledge.

## 1. Introduction

Palliative care is the active, total care to improve the quality of life of patients, whose disease is unresponsive to curative treatment. This interdisciplinary approach addresses patient, family, and community needs, affirms life, considers death and dying as a normal process, and aims to achieve, support, preserve, and improve, as much as possible, the best quality of life until death [1]. As people are living longer with an increased disease burden, many patients will need primary, secondary, and specialist palliative care. Palliative care is a public necessity [2,3] and a human right [4], meaning that people have the right to live free of cruel and degrading side-effects of treatment, the right to nondiscrimination and equality/equal access, as well as the right to information. 

Every year, 1.5 million Americans die in palliative care units (60%), and there are many others who meet the admissibility criteria for these units but are not admitted because of stigma, fears, and misinformation linked to end-of-life care. Patients with an advanced disease receive palliative care very late in their illness course, probably due to fear and stigma attached to palliative care [5]. As palliative care develops further, patients and their families will benefit earlier on from such care.

There is evidence in the literature about the benefits of discussing and dispelling misconceptions about palliative care, as these affect the care and quality of life of people in need of such care. Access to palliative care is also considered an important issue to reduce the global cancer burden by the year 2025 [6] and improve the quality of life of patients and families. The Union for International Cancer Control published a declaration with the goals to be achieved by 2025, one of which is to dispel myths and misconceptions about the disease and reduce the associated stigma [6].

Several studies have assessed knowledge about palliative care in healthcare (HCPs) and nonhealthcare professionals (NHCPs). Lack of such knowledge is associated with myths and misconceptions, which lead to low referral rates and misuse of such services. There is a low level of knowledge among the general population [7,8,9,10]. There is also poor training in palliative care among HCPs. This is considered one of the biggest barriers to promoting the quality of palliative care [11,12] and, therefore, inclusion in the curricula of HCPs is indispensable [13].

Several authors have highlighted the common myths, misconceptions, and superstitions attached to palliative care in several countries, which are linked to poor knowledge by patients, families, and HCPs (especially nurses and doctors) [5,14,15,16,17,18,19,20]. Common myths frequently associate palliative care with a loss of hope for patients, who are merely waiting to die, when nothing more can be done. Consider palliative care for cancer: it is normal to expect and tolerate pain at the end of life. Other misconceptions are related to symptom control, such as the use of oxygen to relieve dyspnea and prolong a patient’s life. There are also misconceptions regarding the management of pain and other symptoms, particularly those related to morphine, the inevitability and normality of pain, as well as nutrition and hydration in palliative care [14,16,17]. Other myths refer to communication, such as the conspiracy of silence or talking about the prognosis and discussing the place of death [5,14,15,16,17,18,19,20].

Myths and misconceptions have been recognized as a problem for patients and professionals by several palliative care national organizations. Information campaigns have been studied to address palliative care literacy. Improving knowledge about palliative care can include the development of a web page and short videos [21] for adults and children, emphasizing the meaning of palliative care for all diseases [22]. Regarding HCPs, content on basic palliative care should be mandatory in undergraduate nursing school curricula, with more indepth content in postgraduate curricula. In clinical practice, palliative care education should be reinforced through support networks for specialist nurses to model excellence in palliative care [23].

To increase health literacy in palliative care and address myths and misconceptions, the first step is identifying the misconceptions and myths among patients, families, HCPs, and the general public. In Portugal, so far, there has been no study of palliative care “knowledge” based on common myths. The level of knowledge of HCPs compared to the general population is also unknown. Therefore, it is difficult to design campaigns to improve palliative care knowledge addressing specific targets. 

The aims of this study are (1) to evaluate the knowledge of HCPs and other professionals in Portugal about palliative care; (2) to identify the terms associated with palliative care used by HCPs and NHCPs in Portugal; (3) to correlate knowledge about palliative care of HCPs and NHCPs in Portugal; (4) to analyze whether certain participant characteristics (age, gender, working in palliative care, relative receiving palliative care) are associated with different levels of knowledge about palliative care.

We hypothesize that HCPs would reveal better knowledge compared to the general public, and HCPs working in palliative care would reveal better knowledge compared to other professionals. 

## 2. Methodology 

### 2.1. Study Design

This was an online cross-sectional study. 

### 2.2. Population and Sample

We used convenience and snowballing sampling techniques, with Portuguese aged 18 years or older, who answered a questionnaire made available online through social media platforms (Facebook Menlo Park, CA, USA, and LinkedIn, Mountain View, CA, USA) during April 2018. The study was first advertised on the Facebook page and then disseminated throughout the web. 

Apart from the age limit of 18 and over, no specific inclusion/exclusion criteria were set for this research. As the online survey could be sent to participants under the age limit, an initial question asked whether the respondent was aged 18 years and over, determining the continuation of the questionnaire. On average, 5–7 min was required for questionnaire completion.

The sample included 592 participants, of which 152 were HCPs (namely, doctors, nurses, physiotherapists, and occupational therapists), and 440 had other professional activities (e.g., managers, economists, teachers, cashiers, electricians, hairdressers, programmers, secretaries). Participants whose professions are not in healthcare are referred to as NHCPs.

### 2.3. Instruments

Given the problem under study, we chose to apply a self-administered questionnaire consisting of three parts:Sociodemographic and family data, namely, profession, age, gender, prior history of a family member hospitalized in palliative care, work experience in palliative care;An open-ended question about what terms participants associate with palliative care;An instrument consisting of 26 statements to assess the knowledge about the goals and purposes of palliative care. The 26 dichotomous (truth or false) questions covered the basics of palliative care, namely, the principles of palliative care, communication, symptom control, family support, teamwork, and organization of care. Incorrect and correct answers were given a score of zero and one, respectively, so totals ranged from 0 to 26, and 13 was the instrument’s median value.

Construction of question statements followed four steps: reviewing descriptions in the literature about the theme; selecting assertions that seemed, to us, most significant; consulting guidelines in the National and European plans for palliative care; considering myths most often indicated by palliative care units/Hospices online pages in several countries: Canada, United States of America, Spain, Portugal, Brazil, and Australia.

Content validity was performed by palliative care experts using a focus group. The focus group consisted of five doctors and five nurses with training and experience in palliative care. We asked experts to indicate whether or not each of the proposed statements were relevant to assess the content of the instrument and/or to revise items if necessary. The expert panel evaluated the instrument’s wording and item allocation. Following that process, all 26 items were retained with high agreement (>80%).

For qualitative face validity, 12 HCPs and 12 NHCPs were asked to express their understanding of the 26 statements. They were also asked about the level of difficulty, fitness, and ambiguity of the items. The items were edited according to this group’s recommendations. There were only suggestions related to language, not content, and there was no difficulty in understanding the items.

### 2.4. Formal and Ethical Procedures

The study was conducted in accordance with the Declaration of Helsinki, and the protocol was approved by the Ethics Committee of Nursing School of Coimbra (project identification n°428-06-2017). Participation in the study was completely voluntary and anonymous. Implicit consent for the project was assumed when study participants completed the survey. The participants received no compensation.

After the participants agreed to participate in the study, the data collected about the respondents were stored in a secure place, inaccessible by unauthorized persons.

### 2.5. Data Processing

The analysis of quantitative variables (i.e., expressed in numbers) was performed by calculating the mean, standard deviation, median, minimum and maximum values. The analysis of qualitative variables (i.e., not expressed numerically) was conducted by calculating the number and percentage of occurrences of each answer.

As the Shapiro–Wilk test indicated the absence of a normal distribution, nonparametric tests were used. Qualitative variables between groups were compared using the chi-square test (with Yates correction for 2 × 2 tables). Quantitative variables between the two groups were compared using the Mann–Whitney *U*-test.

Analyses were performed with the statistical software SPSS v.25 (IBM Corp., Armonk, NY, USA) and, in all cases, with a significance level of 0.05.

In order to analyze the data from the open-response question, we resorted to content analysis, as defined by Bardin [24].

## 3. Results

The 592 participants had a mean age of 31.3 ± 11.1 years (ranging from 18 to 62 years), most were female, did not work in health care, and had no relatives in palliative care. Of the 152 participants working in healthcare, only 18 worked in palliative care (Table 1).

In 24 of the 26 statements, there was a statistically significant difference in correct answers between HCPs and NHCPs, where the former provided the highest percentage of correct questions (see Table 2). In only two statements did both groups reveal the same level of knowledge, namely, the statements regarding hope for palliative care patients and associating palliative care and patients with a life expectancy shorter than 6 months. 

Among the statements with a percentage of wrong answers greater than 50%, we highlight the following: palliative care treats pain with addictive drugs; palliative care is for everyone regardless of age; patients in need of palliative care are dying (NHCPs); we can only receive palliative care at the hospital (NHCPs); only the doctor can refer for palliative care (NHCPs); other treatments have to be stopped in order to access palliative care; palliative care is for patients who have family members to help with care (NHCPs).

Knowledge about palliative care is statistically better (U = 10,605.500; *p* < 0.05) among HCPs (21.1 ± 2.7) when compared to the general population (15.5 ± 5.2). While there is no gender difference among the general population, the same does not happen among HCPs, where women have more knowledge about palliative care (see Table 3). HCPs working in palliative care have, on average, a similar level of knowledge compared to those who do not work in that field (*p* > 0.05).

We also found that older participants have better levels of knowledge, both among HCPs (rs = 0.216; *p* < 0.001) and the general population (rs = 117; *p* > 0.01).

Regarding the terms associated with palliative care (see Figure 1), the term most frequently mentioned by NHCPs was chronic and progressive disease (76). Most of the terms indicated by NHCPs had a positive connotation, for instance, specialized care (39), symptom control (36), quality of life promotion (35), and affection/love (28). Nevertheless, participants associated “nothing else can be done” (4) with palliative care. HCPs used mostly positive terms in association with palliative care: quality of life promotion (29), comfort (26), caring (19), dignity (17), and humanization (10). The terms mentioned by HCPs mostly reflect aspects related to professional clinical practice. We also note the common terms used both by HCPs and NHCPs: quality of life promotion, pain and pain relief, symptom control, communication, love, hope, comfort, dependency, compassion, and dignity. 

## 4. Discussion

A total of 152 HCPs and 440 NHCPs participated in the study, representing a working-age adult sample. HCPs were, on average, older than NHCPs, and the difference was statistically significant for both sexes.

In the entire sample, on average, participants had a level of knowledge about palliative care higher than the instrument’s median value (13). The population of NHCPs had, on average, a reasonable level of knowledge about palliative care. These data are not consistent with other studies [7,8,9,10,25,26]. Despite these figures, on average, the general population failed 10 questions, so improving knowledge is important as it is associated with increased receptivity to these services [27,28].

One-third of nonhealthcare participants indicated “nothing else can be done” as one of the misconceptions about palliative care, congruent with myths previously identified in the literature [14,18]. This was also one of the terms associated with palliative care by NHCPs. In addition to this nothing-can-be-done aspect, almost half of the participants considered that palliative care is for patients with a life expectancy of less than 6 months, as reported by some authors when mentioning that these patients will die soon [14,17,19] or are waiting to die [14,17,19,20]; the latter was indicated by the majority of the general population surveyed. Another misconception was that palliative care is only for people with cancer [15,16], an idea reiterated by a minority of HCPs and almost half of the study’s total sample.

This misconception by the general public might hinder equal access to palliative care of people suffering from other complex conditions of frailty. Palliative care should be available to all those in need regardless of age, diagnosis, or care setting [1,15,29]; therefore, frailty is a clinical syndrome that should be recognized and flagged by the general population. Increasing the knowledge of palliative care associated with frailty will prepare people to be on the alert and report symptoms of frailty in older people (unintentional weight loss, exhaustion, weakness/grip strength, slow walking speed, and low physical activity) [30]. Recognizing severe frailty and early referral to palliative care can reduce pain and discomfort associated with symptoms and enhance the quality of life [30]. 

While hope was one of the terms associated with palliative care by both HCPs (5) and the general population (2), we found that both groups believed there is no hope for people in palliative care [14,17,19,20]. This misconception may indicate that the participants associated palliative care with the hope of living longer rather than living with hope [19,20].

More than half the participants from both subsamples opined that palliative care is not intended for all people regardless of age, which is one of the misconceptions pointed out by several authors about the scope of palliative care [19,20]. On the other hand, other authors [20] have pointed out that the general population thinks palliative care is for those who do not need highly differentiated care. 

Palliative care can be provided in a hospital setting and home care setting [15,16]; however, more than half of NHCP participants wrongly considered this type of care as only administered in a hospital setting. Dependent and bedridden patients were terms associated with palliative care by the nonhealthcare population, in contrast with other studies [5] that have indicated one of the misconceptions about palliative care is that it is only for bedridden patients.

Half of the NHCPs understood that palliative care is for patients who have family members to help with care. While a family presence and participation in caring for palliative care users are important, they are not requirements [20].

Pain, pain relief, suffering, and relief of suffering were also terms associated with palliative care by both groups of participants, HCPs (8, 6, 1, and 7, respectively) and NHCPs (16, 5, 38, and 12, respectively). There were also some misunderstandings related to pain management, including morphine as a very strong pain reliever suitable only in the terminal phase [14].

One indicator that most participants failed was related to pain management, namely, “palliative care is about pain with addictive drugs”, where 55.9% of HCPs and 84.3% of NHCPs indicated the incorrect answer. Previously, some authors [14,15,16,17] have stated there is a misconception that morphine causes respiratory depression and may accelerate death. This last reference, that is, “morphine in palliative care accelerates death”, was reported by a small percentage of HCPs and one-third of NHCPs. Pain is normal and unavoidable [14,16] and this aspect was reported by both healthcare professionals and nonhealthcare participants.

The idea that in order to have access to palliative care, other treatments must be stopped was mentioned by Valassi [18] as a wrong idea. This goes against what was highlighted by the majority of HCPs and less than half of NHCPs in our study. Notably, one-third of nonhealthcare participants agreed that patients in palliative care who stop eating would die of hunger [14,16].

Studies in several countries have reported that the terms most associated with palliative care are quality of life and relief from suffering [7,9,10,31], comfort, pain relief, and dignity [10]. In this study, notably, the terms most referenced by HCPs were quality of life promotion, comfort, and caring, while NHCPs most referred to chronic and progressive disease, specialized care, and symptom control. Common terms linked to palliative care by participants with different professional backgrounds suggest that generalized knowledge has a similar impact upon HCPs and NHCPs. Nevertheless, the HCPs used specific terms, such as humanization, which link palliative care to professional practice. 

Palliative care improves patient and family quality of life [17], as reported by 29 HCPs and 35 NHCPs, and focuses on comfort, dignity, and emotional support of both the patient and their family [5]. These three terms were referenced, respectively, by HCPs and NHCPs. In a 2010 study, palliative care was associated with end-of-life and terminal care [32], where end-of-life was referenced zero times by HCPs and 16 times by NHCPs.

The literature [10,25,33,34] reports that women have a higher level of knowledge about palliative care. However, in this study and [7], women in the general population did not exhibit more knowledge in this area. The same was not true concerning gender among HCPs: women did have more knowledge about palliative care compared to men.

Regarding age, we found that older participants had higher levels of knowledge among both HCPs and NHCPs. This result is similar to other studies [17,25], but contrary to those of Alkhudairi [7].

Participants with no experience of relatives in hospice care have a better knowledge of palliative care. This is contrary to other studies, where participants who knew someone who needed palliative care had better knowledge [10,25]. This result underlines the need for professionals to reflect on the information given to family caregivers and relatives in Portuguese palliative care units. Probably a new approach is needed to inform and clarify the philosophy and goals of palliative care in clinical settings in order to increase the knowledge of NHCPs. It was hypothesized that HCPs working in palliative care would reveal better knowledge compared to other professionals. This hypothesis was not confirmed, revealing misconceptions and myths among professionals who were expected to be specialists. The results of this study emphasize the need to improve knowledge in palliative care, not only in the general population but also among HCPs. Therefore, the education of professionals, especially palliative care professionals, should be guided towards conferring expertise and knowledge, leading to the specialization of these professionals. Attitudes and perceptions about palliative care should also be examined in order to optimize palliative care for people who suffer from a life-threatening and terminal illness. There should be education and training at different levels so professionals can obtain specific knowledge and develop a set of general and specialized competencies in palliative care. On the one hand, general education on palliative care should be established at an undergraduate level for all students undertaking their primary education in any health care discipline, as well as in public health education. On the other hand, a postgraduate level on palliative care should be encouraged for those HCPs who have, or will have, palliative care as the main focus of their work [35,36]. Finally, NHCPs need to be able to talk about life, aging, and the process of living with life-threatening diseases, dying, and health. Through community learning approaches and activities, raising societal awareness about living at older ages, palliative care, dying, and death can also contribute towards including palliative care as an inherent component of care. 

### Study Strengths and Limitations

There were several limitations to this study. The most relevant limitations were the sampling method, sample size, the difference between the two sample groups, and how the data collection instrument was applied. Participants in this study were mostly young adults, which might bias results if age affects knowledge levels. This limitation of the sampling method is probably associated with the use of social media to distribute the questionnaire. In future studies, a different sampling method should be considered to guarantee representativeness. In addition, the sample was limited to internet and digital devices users. Therefore, we acknowledge the results might be different if the questionnaire were applied in rural areas addressing an older population in face-to-face interviews. Although the questionnaire was built based on the literature and was subject to content validation by experts in the field of palliative care, the internal consistency was not tested. This study had, as its main focus, the knowledge about palliative care that can determine misuse of these services, namely, nonreferral to palliative care units. However, attitudes and perceptions about palliative care should also be examined in order to optimize palliative care for people who suffer from a life-threatening and terminal illness.

Despite the limitations, it is noteworthy that the results allow us to realize that there is a need to inform and train HCPs and NHCPs about palliative care. The strengths of the study are also determined by the methodology used for instrument distribution and participant selection. The use of social networks and online forms to collect information allowed us to reduce time and other resources usually devoted to participant selection, data collection and coding, and, therefore, to accelerate the research process and reduce costs.

## 5. Conclusions

In sum, our findings clearly show gaps in the knowledge of healthcare workers in the area of palliative care. The findings also suggest low public awareness about palliative care and its benefits. Misperceptions are, therefore, still relatively prevalent. Our findings are in line with much of the existing research [26,28]. The term most frequently mentioned in association with palliative care among NHCPs was chronic and progressive disease, and, among HCPs, quality-of-life promotion. Women, the elderly, and HCPs revealed a higher level of knowledge regarding palliative care. Participants with family members hospitalized in palliative care did not reveal greater knowledge of palliative care compared to those without such experience. Similarly, HCPs working in palliative care did not reveal better knowledge on the subject.

The results of this study emphasize the need to improve knowledge in palliative care, not only in the general population but also among HCPs. As no differences in knowledge were observed between palliative care professionals and HCPs, other variables, such as attitude, should be addressed in further studies. A multi-center study in collaboration with researchers in European and non-European countries that focused on the factors influencing palliative care knowledge would increase our comprehension of palliative care myths and misconceptions. 

Improving knowledge and enabling HCPs will promote the demystification of these healthcare services, improve access and quality of life for users and their families, and bring us closer to meeting the objectives set by the World Health Organization and international associations in this area. 

## Figures and Tables

**Figure 1 ijerph-17-04630-f001:**
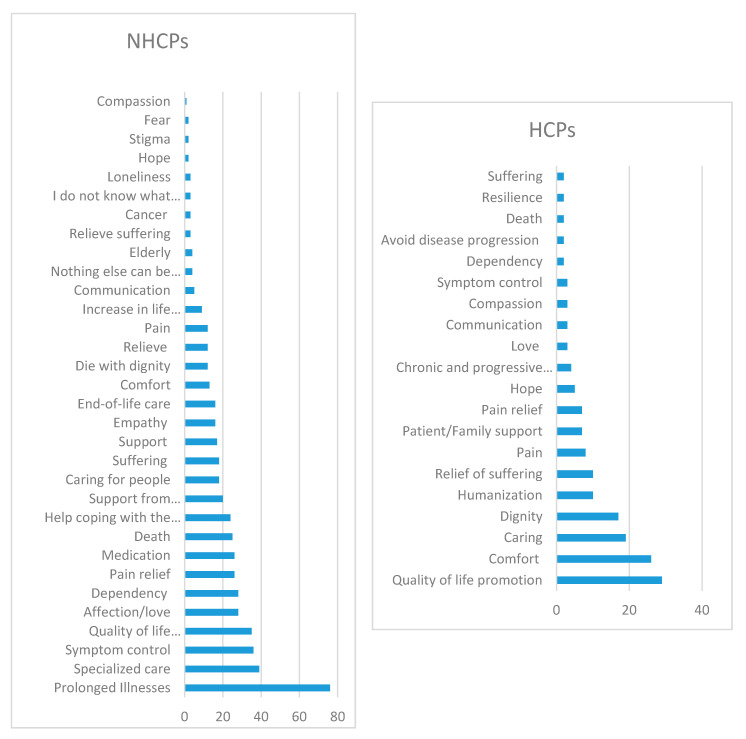
Terms associated with palliative care.

**Table 1 ijerph-17-04630-t001:** Distribution of sociodemographic, professional, and family variables among respondents.

Variables		*N*	%
Gender	Female	487	82.3
	Male	105	17.7
Has any member of your family been hospitalized in Palliative Care?	Not	463	78.2
	Yes	129	21.8
Profession	Health professional	152	25.7
	Non-health professional	440	74.3
If you are a healthcare professional, have you worked/work in Palliative Care?	Not	134	88.2
	Yes	18	11.8

**Table 2 ijerph-17-04630-t002:** Results of applying the chi-square test to the knowledge of healthcare professionals (HCPs) and nonhealthcare professionals (NHCPs).

Indicators	HCPs		NHCPs		*X* ^2^	*p*
	Correct	Wrong	Correct	Wrong		
	*N* (%)	*N* (%)	*N* (%)	*N* (%)		
Palliative care is for cancer patients only *	137 (90.1)	15 (9.9)	297 (67.5)	143 (32.5)	28.428	0.000
Palliative care only treats pain with addictive drugs *	67 (44.1)	85 (55.9)	69 (15.7)	371 (84.3)	49.891	0.000
Palliative care is for everyone regardless of age	67 (44.1)	85 (55.9)	115 (26.1)	325 (73.9)	16.249	0.000
Only end-of-life patients need palliative care *	142 (93.4)	10 (6.6)	375 (85.2)	65 (14.8)	6.135	0.013
Palliative care accelerates death *	126 (82.9)	26 (17.1)	256 (58.2)	184 (41.8)	29.073	0.000
Patients in need of palliative care are in the process of dying *	114 (75.0	38 (25.0)	145 (33.0)	295 (67.0)	79.455	0.000
Pain in palliative care is normal and inevitable *	126 (82.9)	26 (17.1)	318 (72.3)	122 (27.7)	6.243	0.012
Palliative care professionals promote euthanasia *	150 (98.7)	2 (1.3)	325 (73.9)	115 (26.1)	42.338	0.000
We can only receive palliative care at the hospital *	128 (84.2)	24 (15.8)	175 (39.8)	265 (60.2)	87.516	0.000
Palliative care per se does not increase health costs	130 (85.5)	22 (14.5)	268 (60.9)	172 (39.1)	29.968	0.000
Pain is part of death in palliative care *	144 (94.7)	8 (5.3)	348 (79.1)	92 (20.9)	18.601	0.000
Palliative care patients who stop eating starve *	137 (90.1)	15 (9.9)	303 (68.9)	137 (31.1)	25.675	0.000
There is no hope for patients receiving palliative care *	138 (90.8)	14 (9.2)	381 (86.6)	59 (13.4)	1.474	0.225
The doctor who sends patients for palliative care gives up on them *	137 (90.1)	15 (9.9)	267 (60.7)	173 (39.3)	43.862	0.000
Morphine in palliative care is only for the dying patients *	144 (94.7)	8 (5.3)	296 (67.3)	144 (32.7)	43.226	0.000
Morphine in palliative care accelerates death *	143 (94.1)	9 (5.9)	323 (73.4)	117 (26.6)	27.589	0.000
Palliative care means “there’s nothing left to do” *	137 (90.1)	15 (9.9)	303 (68.9)	137 (31.1)	25.675	0.000
Palliative care focuses only on the death process *	144 (94.7)	8 (5.3)	363 (82.5)	77 (17.5)	12.780	0.000
In palliative care, patients are frequently put to sleep	144 (94.7)	8 (5.3)	325 (73.9)	115 (26.1)	28.649	0.000
Palliative care is only useful for pain control *	147 (96.7)	5 (3.3)	321 (73.0)	119 (27.0)	37.082	0.000
Only the doctor can refer to palliative care *	113 (74.3)	39 (25.7)	130 (29.5)	310 (70.5)	91.845	0.000
Palliative care is for those who do not need highly differentiated care	132 (86.8)	20 (13.2)	294 (66.8)	146 (33.2)	21.468	0.000
Palliative care is for patients with a life expectancy of fewer than 6 months	80 (52.6)	72 (47.4)	256 (58.2)	184 (41.8)	1.201	0.273
Other treatments have to be stopped in order to access palliative care *	134 (88.2)	18 (11.8)	205 (46.6)	235 (53.4)	78.072	0.000
Palliative care is for patients who have family members to help with care	120 (78.9)	32 (21.1)	220 (50.0)	220 (50.0)	37.547	0.000
Palliative care is for patients who have only a few days of life	150 (98.7)	2 (1.3)	373 (84.8)	67 (15.2)	19.904	0.000

* Wrong statements.

**Table 3 ijerph-17-04630-t003:** Results of the Mann–Whitney U test between the level of palliative care knowledge and gender, relative admitted to palliative care, and working in palliative care.

		HCPs						NHCPs					
		M	SD	Md	M Rank	U	*p*	M	SD	Md	M Rank	U	*p*
Gender	Male	20.15	2.52	20.00	57.37	900.0	0.040	14.60	5.50	15.00	200.30	13,485.0	0.100
	Female	21.26	2.71	22.00	79.23			15.74	5.10	16.00	225.41		
Relative admitted to Palliative Care	Yes	20.72	3.74	22.50	76.47	1205.5	0.998	15.47	4.80	16.00	217.12	17,884.5	0.746
	No	21.17	2.55	22.00	76.50			15.54	5.33	16.00	221.64		
Works at Palliative Care	Yes (18)	21.27	2.73	22.00	79.36	1154.5	0.766						
	No (134)	21.10	2.71	22.00	76.12								

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
