# Peer review of "Knowledge and Myths about Palliative Care among the General Public and Health Care Professionals in Portugal"

_ijerph, 2020, doi:10.3390/ijerph17134630_

Round 1
Reviewer 1 Report
I thank the authors for the revision for their manuscript.
My concerns raised have been answered properly.
Author Response
Reviewer 1:
> I thank the authors for the revision for their manuscript.
My concerns raised have been answered properly.
RESPONSE
Thanks. We appreciate your comments.
Reviewer 2 Report
- The answer to study aims should be concluded. Conclusion should be a clear finding and message.
- At least, a new part about education in Conclusion should be moved to the Discussion section. The authors should then discuss the methods and contents of education and the expected effectiveness with some literature references.
- English should be improved by scientific and native check.
- There were some inconsistent expressions such as Mann-Whitney and Mann Whitney or Palliative Care or Palliative care (in Table). Careful writing manner should be required.
- In the text apart from Abstract, the word in the first appearance should be abbreviated after fully spelling it out (e.g. HCPs).
Author Response
Reviewer 2
> English should be improved by scientific and native check.
RESPONSE
Thank you for pointing this out. In this revision, we have made sure to proofread the entire document so that the manuscript will be free of typographical and linguistic errors.
> There were some inconsistent expressions such as Mann-Whitney and Mann Whitney or Palliative Care or Palliative care (in Table). Careful writing manner should be required. In the text apart from Abstract, the word in the first appearance should be abbreviated after fully spelling it out (e.g. HCPs).
RESPONSE
Thank you again for your insightful suggestions. The discourse is redundant and a bit confusing, we reword the text with more precision.
> The answer to study aims should be concluded. Conclusion should be a clear finding and message.
At least, a new part about education in Conclusion should be moved to the Discussion section. The authors should then discuss the methods and contents of education and the expected effectiveness with some literature references.
RESPONSE
We appreciate your comment. We agree with reviewer, so the discussion/conclusion sections were reorganized and improved in order to clarify the mentioned aspects. Please see the boldface sentences.
This manuscript is a resubmission of an earlier submission. The following is a list of the peer review reports and author responses from that submission.
Round 1
Reviewer 1 Report
Review ijerph-640292: Knowledge about Palliative Care, in non-health Care 2 professionals and Health Care Professionals
Really needs to be edited by a native English speaker.
Introduction
The introduction is a collection of single phrases, largely unconnected paragraphs, and incorrect statements, probably due to non-native English use.
Please restructure.
Line 45: change “therapeutic” to “therapies” or “therapeutics”.
Line 45: “They” seems to refer to palliative care, not palliative care philosophy. Also, the use of the pronoun “their” is suboptimal as the people to which it refers is not the main subject.
Line 48: please refrain from using single sentence paragraphs in a scientific article.
49: A capital letter after a comma?
The 3rd paragraph (49-52) describes a paradox. A lot of people are not admitted because of stigma etc, yet, the last sentence suggests limited access (“earlier access”).
Lines 53-55: do misconceptions sustain the number of cancer deaths? That is what is being stated.
I do not understand what is being said in 68-70.
It is commonplace to end the introduction with an introduction of the hypothesis and at least a one or two sentences on what is being studied in the current paper so that the reader knows what to expect. E.g., move the objectives (now under methods) to the introduction, preferably without numbering.
Methods
It does not only seem to be a correlational study.
What are professionals from other areas? The same question for non-healthcare professionals. I assume these people were selected as not being healthcare professionals, but did they have to be a professional in another area? Or were they ‘just’ controls (irrespective of their job)?
Objective c needs to be rewritten. It is not clear what is being said.
Sample
How did you reach these people? Via what types of advertising?
Instrument
There needs to be a more detailed description of the instrument used, especially on the 26 statements (c). I am really glad to see that the authors took the design of their instrument seriously and developed the questions through focus groups, and subsequently also tested the adequacy of the items. Kudos! I believe it would be in place also to include some psychometric data of the instrument. Line 104 “some were scored 1”: when? A statement is either true of false, no?
The second sentence of 2.4 does not belong there (that should be in the preceding section). This is the only place where a paragraph of one or two sentences is okay.
Data processing
Please elaborate on the statistical analyses (what were the dependent variables), when were which tests used? Which software was used.
Results
Were there missing data? How were these treated?
I believe it would be good to add the correct answers. Assuming only healthcare professionals read this paper, there are still quite a few out there giving wrong answers (myself included probably!).
Line161-162. You cannot state this. This is why we have statistics. The two groups in this sample have the same amount of knowledge of this subject.
Please use scientific conventions for Table 3, e.g. lines and all abbreviations in the notes.
Discussion
Please don’t reiterate statistics in the discussion. Reiterate a salient finding by all means, but leave the stats for in the results section.
Author Response
Dear reviewer
Thank you very much for your comments, corrections and suggestions.
Following we present the changes and corrections according to your comments.
there was no opportunity /time to send the article for review in English If you consider the paper for publication, we agree to send it for revise the English.
Introduction The introduction is a collection of single phrases, largely unconnected paragraphs, and incorrect statements, probably due to non-native English use. |
Line 45: change “therapeutic” to “therapies” or “therapeutics”. It was changed to Treatment Palliative care is the active, total care to improve the quality of life of patient whose disease is not responsive to curative treatment. |
Line 45: “They” seems to refer to palliative care, not palliative care philosophy. Also, the use of the pronoun “their” is suboptimal as the people to which it refers is not the main subject. Palliative care is the active, total care to improve the quality of life of patient whose disease is not responsive to curative treatment. |
Line 48: please refrain from using single sentence paragraphs in a scientific article. Changed |
49: A capital letter after a comma? …million Americans die in palliative care units (60%), and there are many others…
|
The 3rd paragraph (49-52) describes a paradox. A lot of people are not admitted because of stigma etc, yet, the last sentence suggests limited access (“earlier access”). |
Lines 53-55: do misconceptions sustain the number of cancer deaths? That is what is being stated. Access to palliative care also is considered an important issue to reduce global cancer burden by the year of 2025 [6] and improve the quality of life of patients and families.
|
Several authors highlight common myths, misconceptions and superstitions attached to palliative care in several countries, linked to poor knowledge of patients, families and health care professionals (especially nurses and doctors), [5] [14] [15] [16] [17] [18] [19] [20]. Common myths frequently associate palliative care to the loss of hope for patients to use when the patient is waiting to die and there is nothing more to be done; consider palliative care only for cancer patients and it is normal to expect and tolerate pain at the end of life. Other misconceptions are related to symptom control, for instance, the use of oxygen relieves dyspnea and prolong patient´s life. Other refer to communication such as the conspiracy of silence or talking about prognosis or discussing the place of death [5] [14] [15] [16] [17] [18] [19] [20].
|
To increase palliative care literacy addressing myths and misconceptions, the first step is to access their presence among patients, families, health care professionals and the general public. In Portugal, so far, there is no study on palliative care knowledge based on common myths. The level of knowledge of health care professionals compared to the general population is also unknown. Therefore, it is difficult to design campaigns to improve palliative care knowledge addressing specific targets. This study aimed to evaluate the knowledge of health professionals and professionals from other areas resident in Portugal about palliative care; identify the words associated with palliative care used by health professionals and non-health care professionals’ resident in Portugal; correlate knowledge about palliative care of health professionals and non-health care professionals’ resident in Portugal; correlate palliative care knowledge and sociodemographic (age, gender, working in palliative care, family in palliative care). It was hypothesized that health care professionals would reveal better knowledge compared to the general public and health care professionals working in palliative care would reveal better knowledge compared to other professionals. |
Methods It does not only seem to be a correlational study. This is an observational cross-sectional study. What are professionals from other areas? The same question for non-healthcare professionals. I assume these people were selected as not being healthcare professionals, but did they have to be a professional in another area? Or were they ‘just’ controls (irrespective of their job)? 592 people of both sexes and aged between 18 and 62 answered, being 152 health professionals (HCP), namely doctors, nurses, physiotherapists, and occupational therapists, and 440 participants have other professional activity (managers, economists, teachers, cashier, electricians, hairdressers, programmer, secretary, etc.). The participants whose professions were different from health, are referred as non-health care professionals (NHCP).
|
Objective c needs to be rewritten. It is not clear what is being said. correlate palliative care knowledge and sociodemographic (age, gender, working in palliative care, relative receiving palliative care) |
Sample How did you reach these people? Via what types of advertising? The study was first advertised through the Facebook of the School of Health Sciences and then spread on the web. |
Instrument There needs to be a more detailed description of the instrument used, especially on the 26 statements (c). I am really glad to see that the authors took the design of their instrument seriously and developed the questions through focus groups, and subsequently also tested the adequacy of the items. Kudos! I believe it would be in place also to include some psychometric data of the instrument. Line 104 “some were scored 1”: when? A statement is either true of false, no? The 26 statements that composed the instrument cover the pillars of palliative care, namely bases and principles of care, communication, symptom control, family support, teamwork and organization of care.
The questions answered wrongly were given a score of 0 and some were scored 1, so the values ranged from 0 to 26, with 13 being the median value of the instrument. |
The second sentence of 2.4 does not belong there (that should be in the preceding section). This is the only place where a paragraph of one or two sentences is okay. Changed |
Data processing Please elaborate on the statistical analyses (what were the dependent variables), when were which tests used? Which software was used. For the data processing, IBM-SPSS®, version 25.0 was used. To determine the relationship between nominal variables, the Chi-Square test, and the Mann Whitney U test were used. Regarding the analysis of the answers to the open-ended question, they were submitted to content analysis according to Bardin |
Results Were there missing data? How were these treated? The instrument was designed to be valid if the participant answers all the questions. Participants were free to withdraw from the study at any time. Therefore there were no missing data in this study. |
I believe it would be good to add the correct answers. Assuming only healthcare professionals read this paper, there are still quite a few out there giving wrong answers (myself included probably!). * - Wrong statments (table2) |
Line161-162. You cannot state this. This is why we have statistics. The two groups in this sample have the same amount of knowledge of this subject. Corrections done. Analyzing the data in Table 2, we found that in 24 of the 26 statements the difference found has statistical significance between the number of statements marked as correct by health care professionals and non-health professionals and the highest percentage of correct questions. belongs to health care professionals. Only two statements reveal the same level of knowledge, referring hope to palliative care patients and palliative care is for patients with a life expectancy fewer than 6 months. |
Please use scientific conventions for Table 3, e.g. lines and all abbreviations in the notes. Changed |
Discussion Please don’t reiterate statistics in the discussion. Reiterate a salient finding by all means, but leave the stats for in the results section. Corrected |

Reviewer 2 Report
The study described the knowledge of palliative care in healthcare and non-healthcare professionals. The results may be some interesting, but many problems were seen in the paper. The comments on the paper were not criticisms and for improvement. The study was on the descriptive basis and did not fully include the analytical data. It was so natural to note that the knowledge levels differed between healthcare and non-healthcare professionals. For instance, if the study analyzed the factors associated with the difference, we can discuss some action and practical approaches to resolve the difference. Shrinking the large differences of knowledge is ideal, but is the necessity is completely justified for real care? What was the goal of the study? At least, the paper should first present the working hypothesis. In carrying out this kind of the study, not only knowledge but also attitude and perception are generally examined. Why were not such methodologies applied to the study? The participants were so young (31 years). The sampling method to guarantee the representativeness for the general finding should be reconsidered. The knowledge levels were affected by age and care experience. The appearance of the paper was problematic. In some parts, one or two sentences were appeared in one paragraph. It was only a simple comment or phenomena. Also, for instance, in large and small capitals of title, an integrity was not seen. Further, for instance, abbreviation and full spelling for healthcare and non-healthcare professionals were complexed throughout the text. Not shredded comments but in-depth discussion was required. Revealing the comprehensive findings (for instance, with the factor or cluster analysis) should be required. There have been so many review articles of knowledge aspects of palliative care. This might be first summarized.Author Response
Dear reviewer
Thank you very much for your comments, corrections and suggestions.
Following we present the changes and corrections according to your comments.
there was no opportunity /time to send the article for review in English If you consider the paper for publication, we agree to send it for revise the English.
The study described the knowledge of palliative care in healthcare and non-healthcare professionals. The results may be some interesting, but many problems were seen in the paper. The comments on the paper were not criticisms and for improvement. The study was on the descriptive basis and did not fully include the analytical data. It was so natural to note that the knowledge levels differed between healthcare and non-healthcare professionals. For instance, if the study analyzed the factors associated with the difference, we can discuss some action and practical approaches to resolve the difference.
This study addressed the knowledge about palliative care based on common myths and misconceptions of health care professionals and general public. The factors studied included some sociodemographic variables. Some results such as the lack of knowledge about palliative care of people whose relative received palliative care
Shrinking the large differences of knowledge is ideal, but is the necessity is completely justified for real care?
The equality and access of care
Lack of knowledge about palliative care is associated with myths and misconceptions which leads to low referral and misutilization of such services.
What was the goal of the study? At least, the paper should first present the working hypothesis.
This study aimed to evaluate the knowledge of health professionals and other professionals resident in Portugal about palliative care; identify the words associated with palliative care used by health professionals and non-health care professionals’ resident in Portugal; correlate knowledge about palliative care of health professionals and non-health care professionals’ resident in Portugal; correlate palliative care knowledge and sociodemographic (age, gender, working in palliative care, relative receiving palliative care).
It was hypothesized that health care professionals would reveal better knowledge compared to the general public and health care professionals working in palliative care would reveal better knowledge compared to other professionals.
In carrying out this kind of the study, not only knowledge but also attitude and perception are generally examined. Why were not such methodologies applied to the study?
The study is focused on common myths and misconceptions related to knowledge. In the future knowledge campaigns should address also perception and attitudes. It is very interesting your suggestion to include attitudes and perceptions. We add this on our study limitations:
This study was mainly focused on knowledge about palliative care which mainly leads do misutilization of this services. However, attitudes and perception about palliative care are also important to be examined in order to optimize palliative care to people who suffer from a life-threatening and terminal illness.
The participants were so young (31 years). The sampling method to guarantee the representativeness for the general finding should be reconsidered. The knowledge levels were affected by age and care experience.
We add this to study limitations
The appearance of the paper was problematic. In some parts, one or two sentences were appeared in one paragraph. It was only a simple comment or phenomena. Also, for instance, in large and small capitals of title, an integrity was not seen. Further, for instance, abbreviation and full spelling for healthcare and non-healthcare professionals were complexed throughout the text. Not shredded comments but in-depth discussion was required. Revealing the comprehensive findings (for instance, with the factor or cluster analysis) should be required. There have been so many review articles of knowledge aspects of palliative care. This might be first summarized.
We include this comments on the revised document

Reviewer 3 Report
In this work authors aimed at investigating the general knowledge about palliative care in a cohort of portuguese people working in an health care setting or not working in the health care. Through a 26-items questionnaire they wanted to test the level of real knowledge about palliative care and they found out that especially people non working in health-related environments lack the principle basics of what palliative care is.
This paper is facing an important topic because in my opinion the awareness of population about which are the options available for terminal diseases is a sign of social and health evolution. And to improve global health we need the population to be aware and educated to different medical terms and health education
Before publication, I would consider however some issues:
- Have the authors thought about social, cultural or religious influence about the lack of knowledge about palliative care? In many social contexts, especially the rural ones, ill people are kept along their own family even if they require important and daily assistance. This fact should be considered.
At this regard I invite authors to perform multicenter studies collaborating with researchers from other countries (european and not) with a different cultural and socio-economic background in order to find out which are the factors to hinder knowledge about palliative care.-Another concept that I would like authors to face, for the sake of the soundness of the article, is the concept of "fraility (Fried et al, 2001) as largely debated between specialists. I think that the term "fraility" is another concept that should be stuck in the mind of normal population, because it is the first step towards the necessity of assistance that could subsequently require palliative cares. The ultimate illness that obliges a patients to recur to palliative care is just the tip of the iceberg of a more complex physical status in which the physical and cognitive status of an old subject is frequently associated with low physical activity, slow walking speed, self-reported exhaustion and weight loss probably due to the association of diverse extrinsic conditions like inflammatory status and co-morbidities like hypertension, diabetes or heart disease, and also of intrinsic conditions like recognized genetic factors (Koch et al. 2013). Since the goal of the paper is to s evaluate knowledge about palliative care in normal social setting, I think that this should be done also taking in account the several steps that bring a person ro recur to palliative care.Check for minor typos along the manuscript (for example in Table 1, the percentual of health professional: the numbers do not match!).
Suggested literature:
-Koch G, Belli L, Giudice T Lo, Lorenzo F Di, Sancesario GM, Sorge R, Bernardini S, Martorana A (2013) Frailty among Alzheimer’s disease patients. CNS Neurol Disord Drug Targets 12, 507-511.
-Fried LP, Tangen CM, Walston J, Newman AB, Hirsch C, Gottdiener J, Seeman T, Tracy R, Kop WJ, Burke G, McBurnie MA; Cardiovascular Health Study Collaborative Research Group. Frailty in older adults: evidence for a phenotype. J Gerontol A Biol Sci Med Sci. 2001 Mar;56(3):M146-56.
Author Response
Dear reviewer
Thank you very much for your comments, corrections and suggestions.
Following we present the changes and corrections according to your comments.
there was no opportunity /time to send the article for review in English If you consider the paper for publication, we agree to send it for revise the English.
In this work authors aimed at investigating the general knowledge about palliative care in a cohort of portuguese people working in an health care setting or not working in the health care. Through a 26-items questionnaire they wanted to test the level of real knowledge about palliative care and they found out that especially people non working in health-related environments lack the principle basics of what palliative care is.
This paper is facing an important topic because in my opinion the awareness of population about which are the options available for terminal diseases is a sign of social and health evolution. And to improve global health we need the population to be aware and educated to different medical terms and health education
Before publication, I would consider however some issues:
- Have the authors thought about social, cultural or religious influence about the lack of knowledge about palliative care? In many social contexts, especially the rural ones, ill people are kept along their own family even if they require important and daily assistance. This fact should be considered.
Thank you for your comment. We will consider this factors in further studies and refer this limitation in the revised document.
At this regard I invite authors to perform multicenter studies collaborating with researchers from other countries (european and not) with a different cultural and socio-economic background in order to find out which are the factors to hinder knowledge about palliative care.-Another concept that I would like authors to face, for the sake of the soundness of the article, is the concept of "fraility (Fried et al, 2001) as largely debated between specialists. I think that the term "fraility" is another concept that should be stuck in the mind of normal population, because it is the first step towards the necessity of assistance that could subsequently require palliative cares. The ultimate illness that obliges a patients to recur to palliative care is just the tip of the iceberg of a more complex physical status in which the physical and cognitive status of an old subject is frequently associated with low physical activity, slow walking speed, self-reported exhaustion and weight loss probably due to the association of diverse extrinsic conditions like inflammatory status and co-morbidities like hypertension, diabetes or heart disease, and also of intrinsic conditions like recognized genetic factors (Koch et al. 2013). Since the goal of the paper is to s evaluate knowledge about palliative care in normal social setting, I think that this should be done also taking in account the several steps that bring a person ro recur to palliative care.
Thank you for your suggestion. We include this analysis in discussion and future recommendations.
Check for minor typos along the manuscript (for example in Table 1, the percentual of health professional: the numbers do not match!).
It is corrected in the revised document

Round 2
Reviewer 1 Report
Review revision 1 ijerph-640292: Knowledge about Palliative Care, in non-health Care 2 professionals and Health Care Professionals
The authors have changed the manuscript considerably according to my suggestions, for which I thank them.
The authors have indicated that the final version will be edited by a native English speaker. I trust that they will do this, as it is indeed necessary.
On page two, the authors state: “…people have the right to live free cruel, inhuman, and degrading treatment…”. I find this a bit confusing. I would assume that treatment is, by definition, not inhuman (nor inhumane). Also, is not cruel as it is performed out of compassion. Finally, treatment, however painful or life/function/appearance-altering, in itself is not degrading. Secondary or side-effects may be experienced as degrading, but treatment in itself is not.
On page 3, line 85: “… children emphasizing palliative care…”. Children here is quite a specific addition. Why is this relevant?
Page 3, line 91-92. “their” should be “its”, and what is being accessed? literacy?
Page 3, line 93: “palliative care knowledge based on common myths”. Maybe the rewrite better conveys the message: “…palliative care ‘knowledge’, i.e. pseudo-knowledge based on common myths…”
Page 4, line 129: change “and some were scored 1” to “and correct statements were scored with 1”.
In my previous review I asked which answers were correct. The authors informed me that the asterisk’s in table 2 designate incorrect statements. When I viewed the table, I wonder if some statements have been forgotten to be designated (e.g. “Palliative care focusses only on the death process” which seems incorrect to me, but has no asterisk). Please check table 2.
Page 10, line 236: I suggest replacing “…be important in…” with “restrain”.
Page 12, lines 307 – 312. I think this is one of the big ‘selling points’ on the importance of increasing knowledge regarding palliative care, and could (should) be mentioned in the abstract and conclusion, if the authors so choose.
Page 13, 323-423: I don’t understand how knowledge leads to misutilization.
Reviewer 2 Report
The report was improved but not much. In particular, the in-depth discussion will be required.
Knowledge levels and scores should be developed in the study. Are the levels and scores associated with the age/employment periods and gender? The correlations between indicators should be demonstrated in the supplemental tables. If the data were relevant, discuss. Introduction had 12 paragraphs and Discussion had 18 paragraphs; it was thus too redundant. In Discussion, the results of the study should be weighted from the viewpoints of novelty. Title should be more specified; e.g., younger care professionals in Portugal. Small and large capitals should be uniformly used in title. What was an inferential analysis? The methods of analysis should be described. Abbreviation or non-abbreviation of Journal should be uniformly used; e.g., see Ref 7 and 9. Or see Tables. Although abbreviation of NHCP and HCP was appeared in Methods (page 4), the full spelling was appeared in several following parts; e.g., see the first paragraph in Discussion. An unnecessary space should be deleted (eg, line 49).